# Secretory leukocyte protease inhibitor (SLPI) promotes cholangiocarcinoma progression via inflammation-associated and vasculogenic mechanisms

Kangsadan Chueajedton[1◉], Chaiwat Chueaiphuk[1◉], Jeranan Inpad[2], Sarawut Kumphune[3], Worasak Kaewkong[1], Damratsamon Surangkul[1], Kanlayanee Sawanyawisuth[4], Ubon Cha'on[4], Suchada Phimsen[1,5]*

1 Department of Biochemistry, Faculty of Medical Science, Naresuan University, Phitsanulok, Thailand, 2 Clinical Research Unit, Faculty of Medicine, Naresuan University, Phitsanulok, Thailand, 3 Biomedical Engineering Institute (BMEI), Chiang Mai University, Chiang Mai, Thailand, 4 Department of Biochemistry, Faculty of Medicine, Khon Kaen University, Khon Kaen, Thailand, 5 Centre of Excellence in Medical Biotechnology (CEMB), Faculty of Medical Science, Naresuan University, Phitsanulok, Thailand

◉ These authors contributed equally to this work.
* suchadaph@nu.ac.th

## Abstract

Cholangiocarcinoma (CCA) is a lethal cancer associated with chronic inflammation caused by *Opisthorchis viverrini* infection, with high prevalence in Thailand. Secretory leukocyte protease inhibitor (SLPI), a serine protease inhibitor involved in inflammation, has recently been identified as an oncogenic factor in several malignancies. However, its role in CCA remains unclear. Here, we demonstrate that SLPI is significantly upregulated during CCA development in both human and hamster-induced tissues. Higher SLPI expression was correlated with poor patient survival based on bioinformatic analyses. SLPI was elevated in highly metastatic CCA cell lines and further inducible by IL-6 stimulation. Overexpression of SLPI enhanced tumorigenic properties, including proliferation, migration, invasion, and in vivo tumor growth. SLPI also increased the activity of matrix metalloproteinases (MMP-2 and MMP-9), promoting metastatic potential. While conditioned media from SLPI-overexpressing cells did not affect angiogenesis, these cells promoted vasculogenic mimicry, with increased expression of VEGFA and VE-cadherin, and decreased N-cadherin. These findings suggest that SLPI promotes cholangiocarcinoma progression through inflammation-associated and vasculogenic mechanisms, highlighting its potential as a candidate molecular target for therapeutic intervention.

## Introduction

Cholangiocarcinoma (CCA) is an epithelial cancer arising from the bile ducts and is one of the most devastating malignancies of the hepatobiliary system. Its global

**Data availability statement:** All relevant data are within the manuscript and its Supporting Information files.

**Funding:** This work was supported by the National Research Council of Thailand (Grant No. N41A660176), awarded to Mr. Kangsadan Chueajedton. The funders had no role in study design, data collection and analysis, decision to publish, or preparation of the manuscript.

**Competing interests:** The authors have declared that no competing interests exist.

**Abbreviations:** ANOVA, Analysis of variance; CCA, Cholangiocarcinoma; CM, Conditioned medium; EMT, Epithelial–mesenchymal transition; FBS, Fetal bovine serum; GAPDH, Glyceraldehyde 3-phosphate dehydrogenase; GEPIA2, Gene Expression Profiling Interactive Analysis 2; GTEx, Genotype-Tissue Expression; HA, Hemagglutinin; HCC, Hepatocellular carcinoma; IHC, Immunohistochemistry; IL-6, Interleukin-6; MMP, matrix metalloproteinase; NDMA, *N*-nitrosodimethylamine; NF-κB, Nuclear factor kappa-light-chain-enhancer of activated B cells; OV, *Opisthorchis viverrini*; PBS, Phosphate-buffered saline; RB, Retinoblastoma; SD, Standard deviation; SLPI, Secretory leukocyte protease inhibitor; STAT3, Signal transducer and activator of transcription 3; TCGA, The Cancer Genome Atlas; VE-cadherin, Vascular endothelial cadherin; VEGFA, Vascular endothelial growth factor A; VM, Vasculogenic mimicry

incidence is increasing, but the disease is particularly endemic in Southeast Asia, especially in northeastern Thailand, where it is a major public health concern. This regional disparity is largely attributed to chronic infection with the liver fluke *Opisthorchis viverrini* (OV), a food-borne parasite transmitted through consumption of raw or undercooked freshwater fish. OV infection triggers a persistent inflammatory response in the bile ducts, primarily mediated by interleukin-6 (IL-6) signaling via the Nuclear Factor kappa B (NF-κB) pathway [1,2]. Over time, this chronic inflammation leads to a cascade of molecular events including oxidative stress, DNA damage, epigenetic modifications, and dysregulation of cellular repair mechanisms, culminating in cholangiocarcinogenesis [3].

A major clinical challenge of CCA is that the disease is typically asymptomatic in early stages and often diagnosed only when metastasis has occurred. Standard treatments such as surgery and chemotherapy offer limited benefits at advanced stages, and survival outcomes remain dismal [1]. Therefore, there is an urgent need to identify novel molecular drivers and therapeutic targets that play a role in the initiation and progression of CCA.

Emerging evidence indicates that the tumor microenvironment plays a critical role in cancer development and progression. In CCA, elements such as inflammatory mediators, immune cells, cytokines, and extracellular matrix components interact to support tumorigenesis and metastatic dissemination [4]. Moreover, tumors require a functional blood supply system for sustained growth. While angiogenesis has traditionally been regarded as the primary mechanism of neovascularization, a process known as vasculogenic mimicry (VM), in which cancer cells form vascular-like networks independently of endothelial cells, has been increasingly recognized as an alternative pathway, particularly in aggressive cancers such as CCA [5].

Among the molecules associated with inflammation and cancer progression, Secretory Leukocyte Protease Inhibitor (SLPI) has garnered attention due to its multifunctional role. SLPI is a low molecular weight, secreted protein and a member of the whey acidic protein (WAP) family. It is produced by epithelial cells and immune cells such as neutrophils and macrophages, where it functions as a serine protease inhibitor with anti-inflammatory and antimicrobial properties [6,7]. SLPI has been shown to modulate immune responses, promote wound healing, and protect tissues from excessive protease activity during inflammation.

Interestingly, recent studies have identified SLPI as an oncogenic factor that is upregulated in several cancer types including lung, ovarian, and pancreatic cancers. It has been implicated in promoting cancer cell proliferation, migration, invasion, and resistance to apoptosis. SLPI may also influence the tumor microenvironment by modulating cytokine expression, remodeling the extracellular matrix, and promoting processes such as epithelial-mesenchymal transition (EMT) and vasculogenic mimicry [8]. However, the role of SLPI in CCA remains largely unexplored.

Given the strong link between chronic inflammation and CCA, and SLPI's established involvement in both inflammatory regulation and cancer progression, we hypothesized that SLPI may act as a key regulator of cholangiocarcinogenesis. In this study, we aimed to elucidate the expression and functional role of SLPI in CCA,

with a particular focus on tumorigenesis, metastatic behavior, and the formation of tumor-associated blood supply systems including angiogenesis and vasculogenic mimicry. Understanding the molecular mechanisms underlying SLPI's contribution to CCA may provide novel insights for early diagnosis and therapeutic interventions in this deadly disease.

## Materials and Methods

### Transcriptomic data sources

To evaluate SLPI mRNA expression in clinical cholangiocarcinoma (CCA) specimens, publicly available transcriptomic datasets were utilized. Raw expression data were retrieved from the Gene Expression Omnibus (GEO) database under accession number GSE53870, which includes transcriptomic profiles from 36 intrahepatic CCA tissue samples and 9 normal intrahepatic bile duct controls. Differential expression analysis and survival correlation were performed using GEPIA2, an interactive online platform integrates TCGA and GTEx RNA sequencing data. This analysis was used to identify the clinical relevance of SLPI expression and to guide subsequent experimental investigations.

### Hamster model of cholangiocarcinogenesis

An inflammation-induced CCA model was established in male golden Syrian hamsters by oral administration of 50 *Opisthorchis viverrini* metacercaria per animal and 12.5 ppm N-nitrosodimethylamine (NDMA) in drinking water for 2 months, followed by monitoring up to 6 months as previously described [9]. Animals were sacrificed at 0, 1, 3, and 6 months for immunohistochemical analyses of SLPI (n = 3 animals per time point). Throughout the experimental period, the health and behavior of animals were monitored at least three times per week by trained personnel. Humane endpoints were predefined and applied to minimize suffering. Animals showing signs of severe distress, such as >20% body weight loss, persistent anorexia for more than 48 hours, lethargy, impaired mobility, abnormal posture, respiratory distress, or ulcerated tumors, were humanely euthanized by intraperitoneal pentobarbital. No unplanned deaths occurred during the study.

All animal experiments were reviewed and approved by the Ethics Committee for Animal Research at Khon Kaen University (AEMDKKU 1/2558) and conducted in strict accordance with institutional guidelines, the ARRIVE guidelines, and the PLOS ONE policy on humane endpoints.

### Immunohistochemistry

Formalin-fixed, paraffin-embedded hamster liver tissues (n = 3 animals per time point) were sectioned, deparaffinized, rehydrated, and subjected to antigen retrieval in Tris-EDTA buffer (pH 9.0) at 121 °C for 3 min. Endogenous peroxidase was quenched with 0.3% $H_2O_2$, and sections were blocked with 1% BSA before overnight incubation with primary antibodies (Supplementary Table S1), followed by HRP-conjugated secondary antibodies. Staining was visualized with DAB and counterstained with hematoxylin. SLPI expression was semi-quantitatively evaluated using the H-score formula: 1 × (% cells 1+) + 2 × (% cells 2+) + 3 × (% cells 3+) [10].

### Cell culture and transfection

MMNK-1 (immortalized cholangiocyte), KKU-213A (low-metastatic CCA cell line) and KKU-213AL5 (high-metastatic CCA cell line) were obtained from the Japanese Collection of Research Bioresources Cell Bank (Osaka, Japan). EA.hy926 (endothelial cell line) was obtained from the American Type Culture Collection. Cells were maintained in DMEM supplemented with 10% FBS and antibiotics at 37 °C in 5% $CO_2$. SLPI overexpression was generated by transfecting MMNK-1 cells with pCMV2-SLPI-HA using Lipofectamine 2000; Mock cells received empty vector. Stable clones were selected with 100 μg/mL hygromycin B, and two clones [SLPI(1) and SLPI(2)] were used for experiments. MMNK-1 cells were used for inflammation modeling and SLPI overexpression experiments, whereas KKU-213A and KKU-213AL5 cells served as

cholangiocarcinoma models with endogenous SLPI expression. EA.hy926 endothelial cells were used exclusively for angiogenesis-related assays.

## Western blot analysis

Cells were lysed in RIPA buffer with protease/phosphatase inhibitors, and protein concentrations were measured by the Bradford assay. Equal amounts of protein (40 µg) were resolved by 12% SDS-PAGE, transferred to PVDF membranes, blocked with 5% skim milk, and probed with primary antibodies (S1 Table) followed by HRP-conjugated secondary antibodies. Bands were visualized by ECL and quantified using ImageQuant 4000 and ImageJ, with β-actin as the loading control.

## Soft agar colony formation assay

Mock, SLPI-overexpressing MMNK-1 cells, and cholangiocarcinoma cell lines were used for soft agar assays. A total of $5\times10^3$ cells suspended in 0.35% agarose gel in complete medium were placed on 0.7% agarose gel in 6-well plates with addition of growth medium, followed by culture at 37 °C in a humidified atmosphere with 5% $CO_2$ for 30 days. Cells were stained with 0.05% w/v crystal violet in 20% methanal at room temperature for 30 min and the number of colonies was counted.

## ELISA

Cells ($4\times10^5$ cells/dish) were cultured in 60-mm dishes with serum-free medium for 48 h, and supernatants were collected, filtered (0.22 µm), and used for assays [11]. For cytosolic protein analysis, cells were washed with cold PBS and lysed in RIPA buffer supplemented with protease inhibitors. Cell lysates were centrifuged at 12,000×g for 15 min at 4 °C, and the supernatants were collected as cytosolic protein fractions. Cytosolic proteins and conditioned media were analyzed using the Quantikine® Human SLPI Immunoassay kit (R&D Systems) following the manufacturer's instructions. Cytosolic SLPI levels reflect intracellular expression, whereas conditioned media were used to assess secreted SLPI. Assays were performed in duplicate (technical replicates), and absorbance was measured at 450 nm with background correction at 570 nm using a microplate reader.

## MTT assay

Mock, SLPI-overexpressing MMNK-1 cells, and CCA cell lines were subjected to MTT assays to assess proliferation. Cells were seeded into 96-well plates ($3\times10^3$ cells/well for MMNK-1 and CCA cells, and $6\times10^3$ cells/well for EA.hy926) and cultured for up to 5 days. MTT reagent (final concentration: 5 mg/mL) was added and incubated at 37˚C for 4 h. After incubation, the formazan was solubilized by DMSO. The optical density of each sample was immediately measured using a microplate reader at 540 nm.

## Wound healing assay

Migration assays were performed using Mock and SLPI-overexpressing MMNK-1 cells, with CCA cell lines included for comparison. Cells ($2\times10^5$ cells/well) were cultured to confluence, scratched with a sterile 200-µl pipette tip, and incubated in fresh media. The images were captured, and wound closure was quantified as the percentage of the initial wound area remaining at each time point using ImageJ.

## Tube formation assay

Matrigel (50 µl/well; Discovery Labware, Inc. USA.) was added to 96-well plates and polymerized at 37 °C for 1 h. EA.hy926 cells ($3\times10^4$ cells/well) were seeded and incubated for 6 h. Tubular-like structures (Tube formation) were

assessed under a light microscope (Olympus, Tokyo, Japan). Endothelial tube formation was quantified using ImageJ software (NIH, USA) by analyzing the number of complete tubular structures per field under identical imaging conditions. Quantification was performed in at least five randomly selected fields per well from three independent experiments.

## Vasculogenic mimicry formation assay

Matrigel-coated 96-well plates were seeded with 8 x $10^4$ cells/well Mock and SLPI(2) cells and incubated at 37 °C in a humidified 5% $CO_2$ and 1% $O_2$ for 4 h. Tubular-like structures were imaged under a light microscope (Olympus, Tokyo, Japan). VM formation was quantified using ImageJ software (NIH, USA) by analyzing the number of tubular-like structures per field under identical imaging conditions. Quantification was performed in at least five randomly selected fields per well from three independent experiments.

## RT-qPCR

Total RNA was extracted from cells using TRIzol reagent (Invitrogen) and converted to cDNA using iScript Reverse Transcription Supermix (Bio-Rad). qPCR was conducted using SYBR Green with primers listed in S2 Table under specific cycling conditions (95 °C for 15 min; 40 cycles of 95 °C for 20 sec, 64 °C for 20 sec, 72 °C for 30 sec). Expression was normalized to GAPDH and calculated using the $2^{-\Delta\Delta Ct}$ method.

## Transwell invasion assay

Invasion assays were performed using Mock and SLPI-overexpressing MMNK-1 cells, with CCA cell lines included as invasive controls. Cell invasion was assessed using 8-µm pore Matrigel-coated transwell chambers. A total of 5x$10^4$ cells/well were seeded in serum-free medium in the upper chamber, with medium containing 10% FBS in the lower chamber as chemoattractant. After 12 h, non-invading cells were removed, and invaded cells were fixed with 4% paraformaldehyde, stained with 0.5% crystal violet, and quantified by solubilizing the dye with DMSO and measuring absorbance at 570 nm.

## Cell adhesion assay

Adhesion assays were performed using Mock and SLPI-overexpressing MMNK-1 cells, with CCA cell lines included as adhesive controls. Cell adhesion was evaluated by seeding 1x$10^4$ cells/well onto 96-well plates pre-coated with 0.4 µg/µL Matrigel and incubated overnight at 4°C. After 1 h incubation at 37 °C, non-adherent cells were removed by washing three times with PBS. Attached cells were quantified using the MTT assay, and absorbance was measured at 540 nm.

## Gelatin zymography

Conditioned media from Mock, SLPI-overexpressing MMNK-1 cells, and CCA cell lines were analyzed for MMP activity. MMP-2 and MMP-9 activities were analyzed by gelatin zymography using serum-free conditioned medium concentrated with 3 kDa filters [12]. Equal protein volumes were mixed with non-reducing buffer, separated on 7.5% SDS–polyacrylamide gels containing 0.1% gelatin, renatured in 2.5% Triton X-100, and incubated overnight at 37 °C in developing buffer. Gels were stained with Coomassie Blue, destained, and band intensities quantified using ImageJ.

## Animal model (Xenograft assay)

Male nude mice (BALB/cAJcl-nu, 6–8 weeks) were injected subcutaneously with 1 × $10^6$ Mock or SLPI-overexpressing cells (n = 9/group). Tumor growth was monitored every 2 days for 48 days, and volumes were calculated as V = ½ (L × W²). At the experimental endpoint or when humane endpoint criteria were met, mice were euthanized, and tumors were excised and weighed. Mice were observed daily for general appearance, body weight, and behavior. Humane endpoint criteria included >15–20% body weight loss, tumor diameter exceeding 1.5 cm, ulceration, impaired movement, or any

 

signs of distress (e.g., hunching, abnormal breathing, or inactivity). Animals meeting these criteria were euthanized using intraperitoneal pentobarbital. All efforts were made to minimize suffering, including provision of soft bedding, controlled environment, and access to food and water ad libitum.

All animal procedures were approved by the Naresuan University Animal Care and Use Committee (NUACUC, Approval No. 6201016) and conducted in compliance with ARRIVE guidelines and institutional animal welfare regulations.

### Statistical analysis

Data are presented as mean ± SD from at least three independent experiments unless otherwise stated. For in vitro assays, "n" denotes independent biological experiments performed on different days; technical replicates (e.g., duplicate or triplicate wells) were averaged within each experiment and are not counted as "n". For animal studies, "n" indicates the number of animals per group. Prior to parametric testing, data distribution was assessed for normality using the Shapiro–Wilk test. For comparisons between two groups, an unpaired Student's t-test was used. For comparisons among three or more groups, one-way ANOVA was applied, followed by Tukey's multiple comparisons test. A p value < 0.05 was considered statistically significant. All analyses were performed using GraphPad Prism version 9 (GraphPad Software, San Diego, CA, USA).

## Results

### SLPI expression in CCA

SLPI mRNA levels were significantly higher in CCA tissues (n = 36) than in normal bile ducts (n = 9) according to GEPIA2, and lower expression correlated with longer disease-free survival (Fig 1a,b). In a hamster model induced by *Opisthorchis viverrini* and NDMA, SLPI expression progressively increased with CCA development, with H-scores rising from 20.00 (control) to 73.33 ± 11.55 (1 month), 116.67 ± 15.28 (3 months), and 166.67 ± 30.55 (6 months), consistent with histopathological changes (Fig 1c,d). ELISA further showed lowest SLPI levels in MMNK-1 cholangiocytes, intermediate levels in KKU-213A, and the highest in metastatic KKU-213AL5 cells (Fig 1e,f). These results indicate that SLPI expression correlates with CCA progression and poor patient prognosis.

### SLPI is upregulated under inflammation mimicking *Opisthorchis viverrini* infection

To examine the effect of inflammation on SLPI, MMNK-1 (cholangiocytes) and KKU-213A (low metastatic CCA) cells were treated with IL-6 (0–10 ng/mL) for 12 or 24 h. Western blot showed dose-dependent increases in NF-κB p65 phosphorylation: in MMNK-1 cells, phosphorylation rose 2.14–4.56 fold at 12 h and remained elevated at 24 h, while KKU-213A cells showed similar trends (Fig 2a-d). SLPI expression was also significantly upregulated in IL-6–treated MMNK-1 cells, with 2.58–5.54 fold increases at 12 h and 1.49–4.16 fold at 24 h (Fig 2e). These results indicate that IL-6-induced inflammation promotes NF-κB activation and SLPI upregulation in cholangiocytes and CCA cells.

Soft agar assays further revealed that IL-6 (10 ng/mL) enhanced anchorage-independent growth, with treated cells forming significantly more colonies than controls (204.33 ± 55.88 vs. 101.56 ± 17.11 for MMNK-1; 172.67 ± 48.04 vs. 75.11 ± 7.86 for KKU-213A) (Fig 2g-i). Thus, inflammatory conditions mimicking *O. viverrini* infection upregulate SLPI and enhance tumorigenicity.

### Establishment of SLPI-overexpressing cholangiocyte cells

To explore the functional role of SLPI, SLPI-overexpressing MMNK-1 cells were generated. Western blot confirmed increased cytosolic SLPI in transfected clones compared with parental and Mock controls: 1.00 (MMNK-1), 1.33 ± 0.39 (Mock), 3.22 ± 0.89 [SLPI(1)], and 5.03 ± 0.29 [SLPI(2)], while endogenous levels were higher in CCA cells (6.87 ± 0.17 in KKU-213A and 8.87 ± 1.23 in KKU-213AL5) (Fig 3a,b). ELISA showed significantly elevated cytosolic and secreted SLPI in

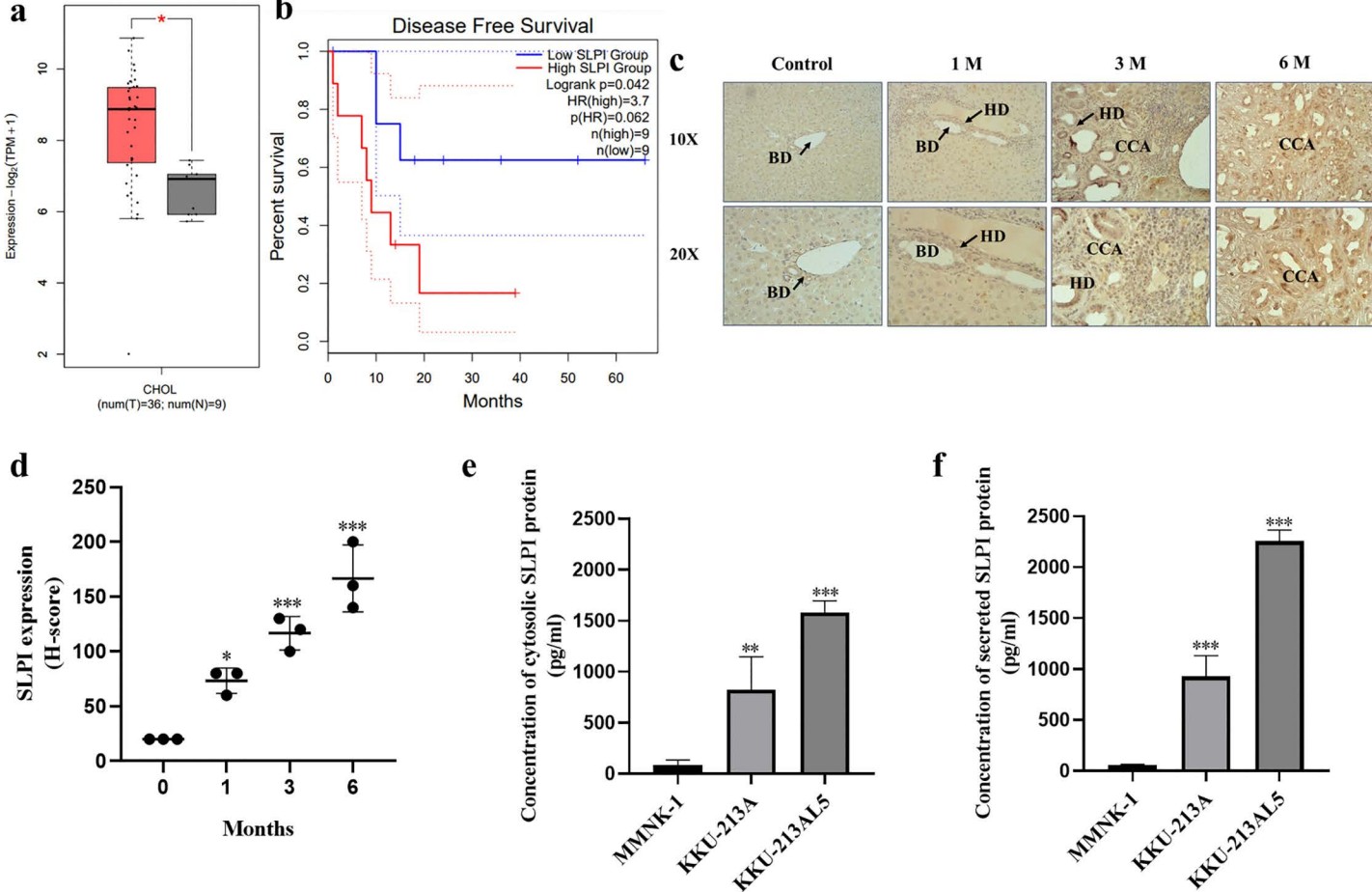

**Fig 1. SLPI expression in human CCA, hamster model, and cell lines. (a)** Relative mRNA expression of SLPI in CCA tissues (n = 36) compared with normal bile duct tissues (n = 9) using GEPIA2 analysis of TCGA dataset. **(b)** Kaplan–Meier analysis showing the correlation between SLPI expression levels and disease-free survival in CCA patients. **(c)** Representative immunological staining of SLPI in bile duct sections of hamster treated with *Opisthorchis viverrini* and NDMA at 0, 1, 3 and 6 months. **(d)** Quantitative analysis of SLPI immunoreactivity expressed as H-scores across each time point (n = 3 per group). **(e)** ELISA quantification of cytosolic SLPI protein levels in MMNK-1, KKU-213A and KKU-213AL5 cell lines. **(f)** ELISA quantification of secreted SLPI protein levels in the corresponding culture supernatants. Data are presented as mean ± SD. Statistical significance was determined using one-way ANOVA followed by Tukey's post hoc test. *p < 0.05, **p < 0.01, ***p < 0.001 compared to MMNK-1 cells.

SLPI-overexpressing clones (349.10 ± 76.44 and 1240.75 ± 44.69 pg/mL for SLPI(1); 594.19 ± 82.24 and 1578.28 ± 145.18 pg/mL for SLPI(2)) compared with MMNK-1 and Mock, but still lower than CCA lines (Fig 3c,d). These findings confirm successful establishment of SLPI-overexpressing cholangiocytes, with expression higher than controls but lower than CCA and metastatic CCA cells.

## SLPI-overexpressing cholangiocytes promote tumorigenicity

To evaluate tumorigenic potential, proliferation of Mock, SLPI(1), SLPI(2), KKU-213A, and KKU-213AL5 cells was compared. SLPI(1) and SLPI(2) showed significantly higher proliferation than Mock, while CCA lines displayed even greater rates than SLPI(2), indicating a positive correlation between SLPI expression and cell growth (Fig 3e). In soft agar assays, SLPI(2) cells formed markedly more colonies than Mock (114.00 ± 12.00 vs. 30.50 ± 9.17), confirming enhanced tumorigenicity in vitro (Fig 4a,b). In xenograft models, mice injected with SLPI(2) cells developed significantly larger tumors by

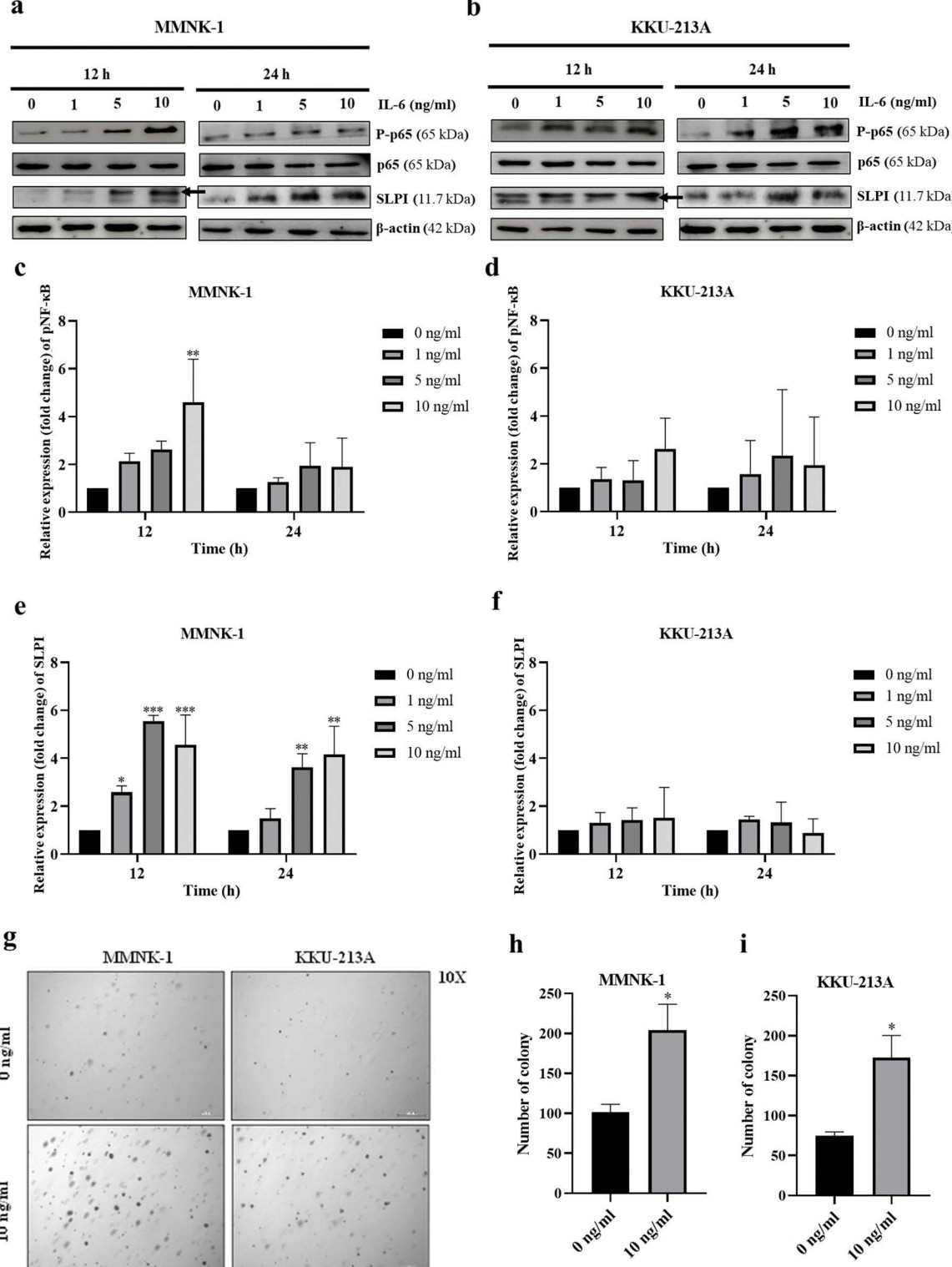

**Fig 2. IL-6-induces inflammation upregulates SLPI expression and promotes tumorigenicity in CCA cells. (a-f)** Western blot analysis of NF-κB p65 phosphorylation and SLPI expression in MMNK-1 and KKU-213A treated with IL-6 (0, 1, 5, and 10 ng/mL) for 12 and 24 hours. Densitometric analyses of pNF-κB and SLPI were normalized to total NF-κB and β-actin, respectively, and expressed as relative expression (fold change) compared

with control. The corresponding original blot images are provided in S1 and S2 Figs. **(g-i)** Quantification of colony formation in soft agar assays following IL-6 (10 ng/mL) treatment in MMNK-1 and KKU-213A cells. Each bar represents the mean ± SD (n = 3). Statistical analysis was performed using unpaired Student's t test. *p < 0.05, **p < 0.01, ***p < 0.001 compared to untreated controls.

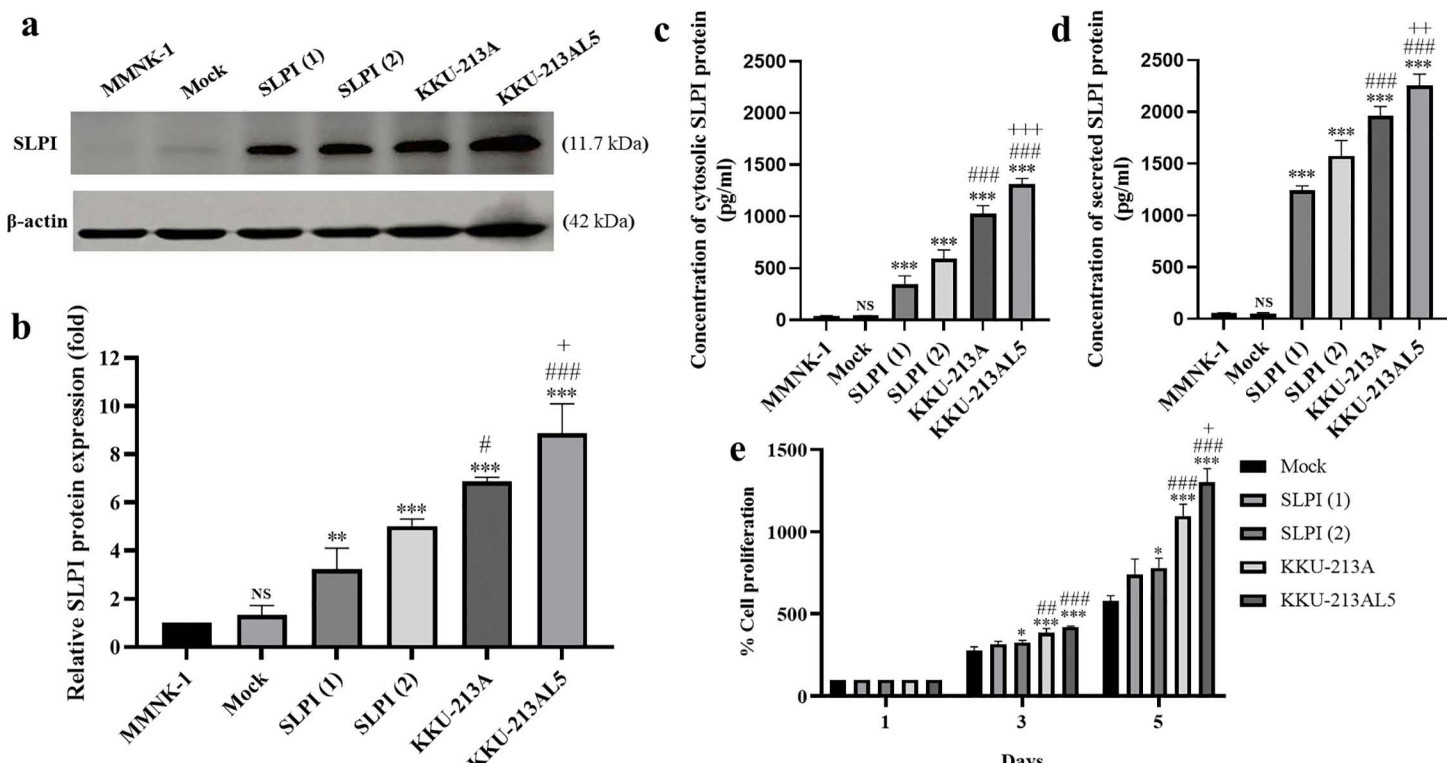

**Fig 3. SLPI expression levels in SLPI-overexpressing MMNK-1 cells and its correlation with cell proliferation. (a, b)** Western blot analysis of cytosolic SLPI expression in MMNK-1, Mock, SLPI(1), SLPI(2), KKU-213A, and KKU-213AL5 cells. The corresponding original blot images are provided in S3 Fig. **(c, d)** ELISA Quantification of cytosolic and secreted SLPI levels in the same cell lines. **(e)** Correlation between SLPI expression and cell proliferation rate across MMNK-1, SLPI-overexpressing, and CCA cell lines. Each bar represents the mean ± SD (n = 3). Statistical analysis was performed using one-way ANOVA followed by Tukey's multiple comparisons test *p < 0.05, **p < 0.01 and ***p < 0.001 vs. MOCK; #p < 0.05, ###p < 0.001 vs. SLPI(2); ++p < 0.01, +++p < 0.001 vs. KKU-213A; NS = no significant vs. MMNK-1.

weight (14.38 ± 5.56 mg) than those injected with Mock (2.12 ± 4.21 mg) (Fig 4c,d), although tumor volume differences were not significant (Fig 4e). Although tumor volume did not differ significantly between groups, SLPI-overexpressing tumors exhibited a significantly greater tumor weight at the experimental endpoint, indicating differences that may not be fully captured by caliper-based volume estimation. Together, these results demonstrate that SLPI overexpression promotes tumorigenic potential in cholangiocytes both in vitro and in vivo.

## SLPI overexpression enhances metastatic potential of cholangiocyte

To assess SLPI's effect on metastatic phenotypes, wound healing, transwell invasion, adhesion assays, and gelatin zymography were performed in SLPI-overexpressing, control, and CCA cells. SLPI(1), SLPI(2), KKU-213A, and KKU-213AL5 cells showed significantly greater migration and invasion than Mock, with CCA lines-especially KKU-213AL5-being most aggressive (Fig 5a–d). Adhesion was also higher in SLPI-overexpressing cells versus Mock, though CCA cells

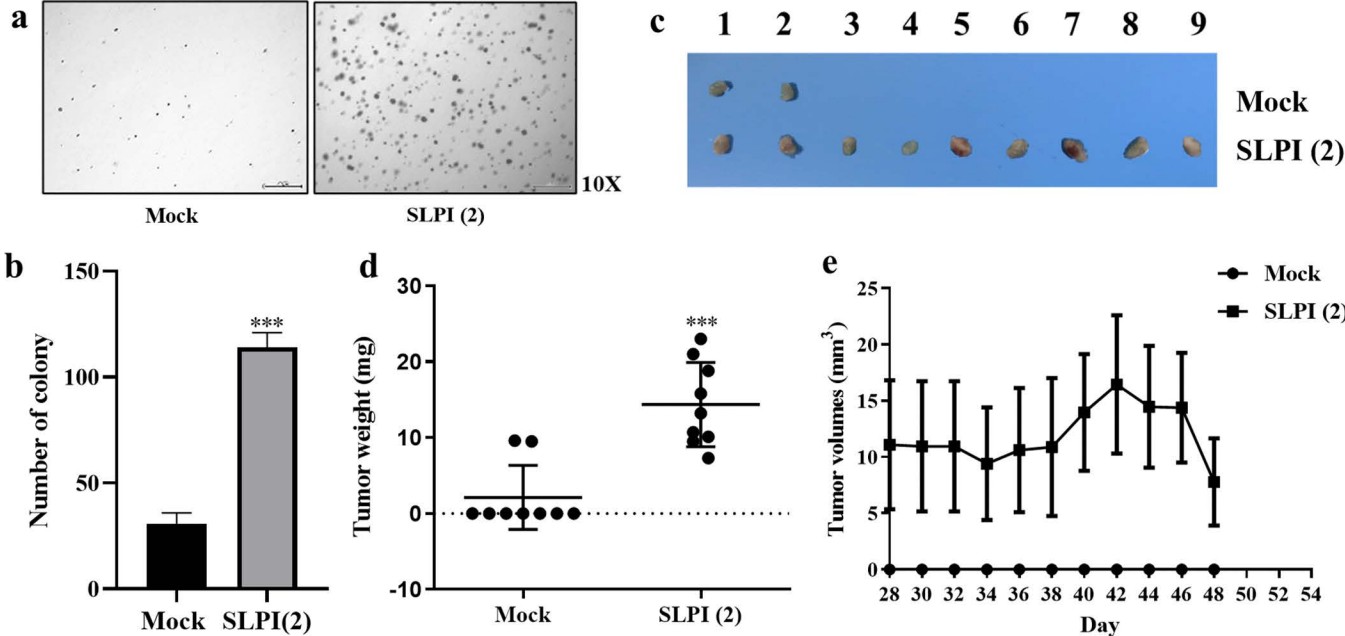

**Fig 4. Tumorigenic potential of SLPI-overexpressing cholangiocyte. (a,b)** Soft agar colony formation assay showing increased anchorage-independent growth in SLPI(2) cells compared to Mock controls. **(c,d)** Tumor weight from xenograft mouse model injected with Mock and SLPI(2) cells (n = 9). **(e)** Tumor volume progression over time in xenograft mice. Data are presented as the mean ± SD. Statistical comparisons were made using unpaired Student's t-test. ***p < 0.001 vs. Mock group.

displayed even greater adhesion (Fig 5e). Gelatin zymography revealed increased pro-MMP2 in SLPI(1) and SLPI(2), but reduced activity in KKU-213A and KKU-213AL5, whereas active MMP2 was elevated only in CCA cells (Fig 5f). Active MMP9 levels were markedly increased in SLPI(1), SLPI(2), KKU-213A, and KKU-213AL5 compared with Mock (Fig 5g). These results indicate that SLPI promotes metastatic potential by enhancing migration, invasion, adhesion, and MMP9 activity.

### SLPI overexpressing enhances vasculogenic mimicry but does not promote angiogenesis

To investigate SLPI's role in tumor blood supply, we assessed its effects on angiogenesis and vasculogenic mimicry (VMF). Conditioned media from Mock and SLPI(2) cells showed no significant differences in endothelial proliferation, migration, or tube formation (Fig 6a-c), indicating no effect on classical angiogenesis. In contrast, VMF was markedly increased in SLPI(2) cells, which formed 279 ± 13 tubular structures versus 46 ± 45 in Mock (Fig 6d). Gene expression analysis confirmed higher SLPI, VEGFA, and VE-cadherin, with reduced N-cadherin, in SLPI(2) compared with Mock (Fig 6e-h). These findings suggest that SLPI supports CCA blood supply through vasculogenic mimicry rather than angiogenesis.

### Discussion

Secretory leukocyte protease inhibitor (SLPI), a serine protease inhibitor, is well known for its anti-inflammatory properties under physiological conditions [13]. However, in several malignancies, SLPI paradoxically plays a tumor-promoting role [14–16]. Our findings revealed that SLPI is significantly upregulated in human cholangiocarcinoma (CCA) tissues, hamster CCA models, and CCA cell lines. Both cytosolic and secreted SLPI levels strongly correlate with advanced CCA stages and are inversely associated with overall survival, highlighting its potential as a prognostic biomarker for CCA. This pattern

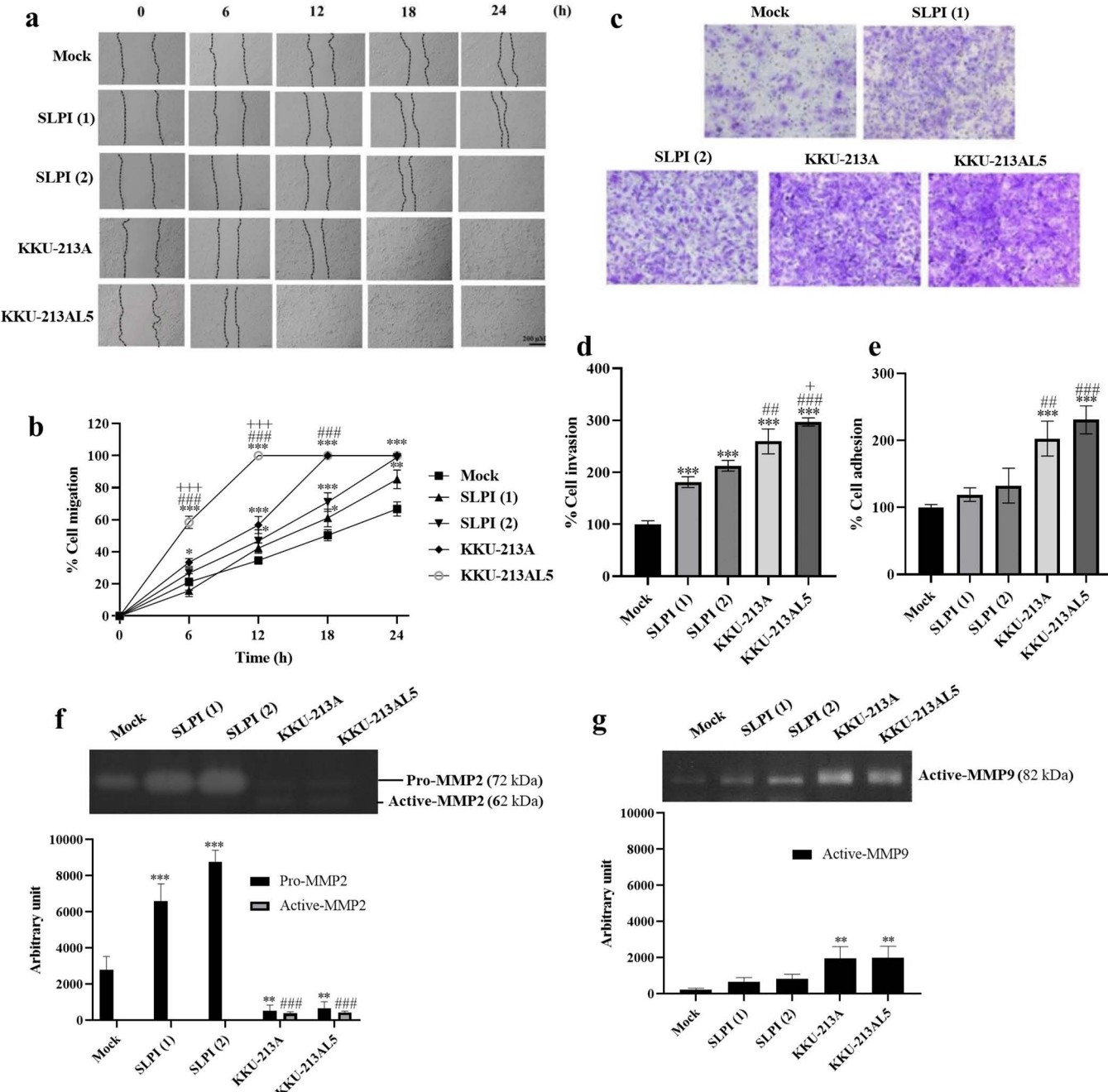

**Fig 5. SLPI overexpression enhances metastatic properties in cholangiocyte. (a)** Representative images from wound healing assay. **(b)** Quantification of relative cell migration. **(c)** Representative images from transwell invasion assay. **(d)** Quantification of relative cell invasion. **(e)** cell adhesion capacity. **(f,g)** Gelatin zymography analysis of MMP-2 and MMP-9 activity in Mock, SLPI(1), SLPI(2), KKU213A and KKU213AL5 cells. The corresponding original gel images are provided in S4 Fig. Data are presented as mean ± SD (n = 3). Statistical comparisons were performed using one-way ANOVA followed by Tukey's multiple comparisons test *p < 0.05, **p < 0.01, ***p < 0.001 vs. Mock; ##p < 0.01, ###p < 0.001 vs. SLPI(2); +p < 0.05·+++p < 0.001 vs. KKU-213A.

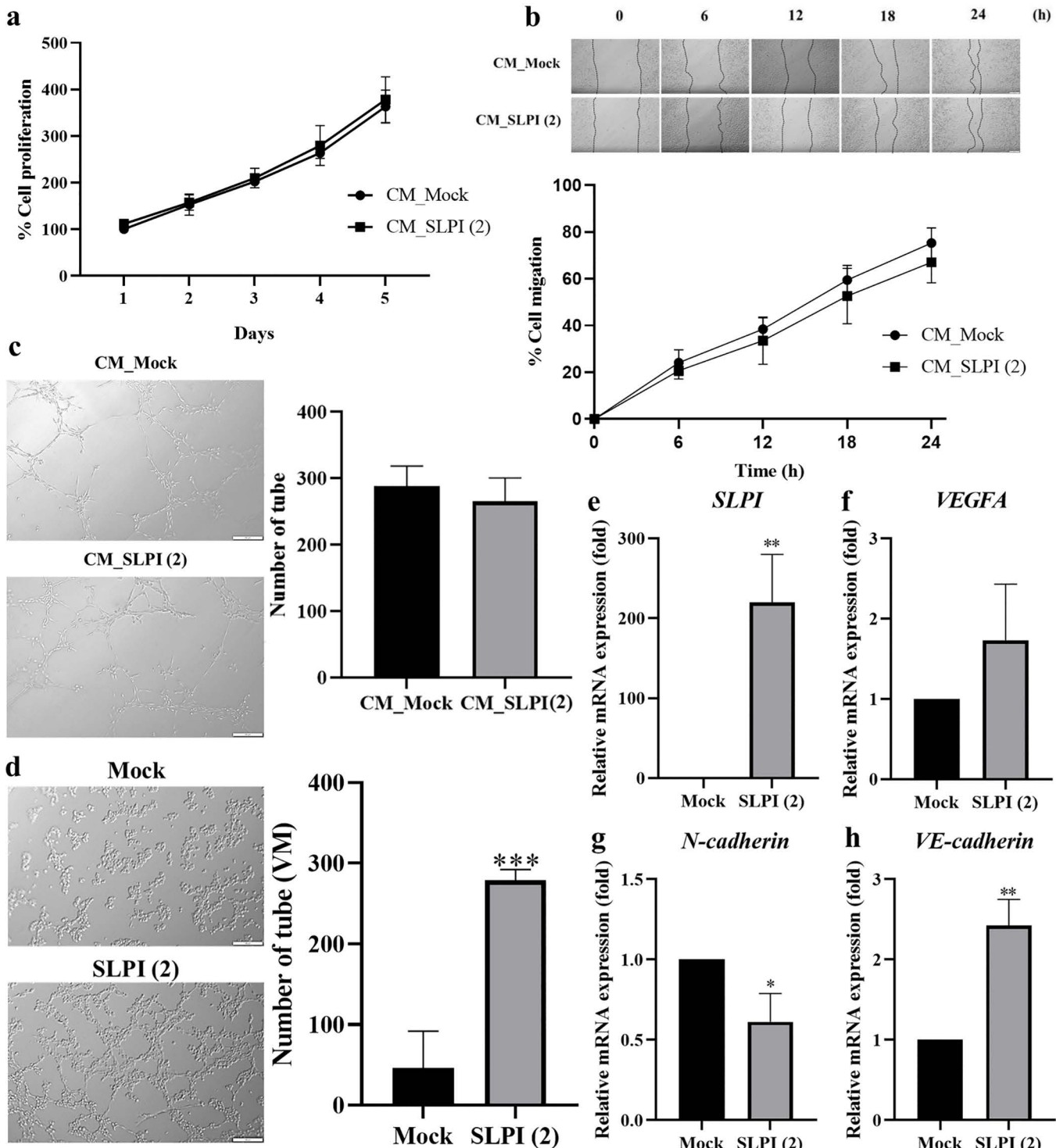

**Fig 6. SLPI overexpression promotes vasculogenic mimicry but not angiogenesis. (a)** Proliferation, (**b**) migration, and (**c**) tube formation of endothelial cells cultured with conditioned media from Mock or SLPI(2) cells. **(d)** VMF ability of SLPI(2) and Mock cells. VM structures were defined as tumor cell–derived tubular-like networks formed in the absence of endothelial cells and were quantified using ImageJ **(e-h)** Relative mRNA expression levels of SLPI, VEGFA, N-cadherin, and VE-cadherin in SLPI(2) and Mock cells. Data are presented as mean ± SD (n = 3). Statistical comparisons were made using unpaired Student's t-test. *p < 0.05, **p < 0.01 vs. Mock.

of elevated SLPI and its correlation with poor prognosis has also been reported in other malignancies, including ovarian and colorectal cancers [17–20]. Conversely, SLPI is notably downregulated in hepatocellular carcinoma [21], suggesting its potential utility in distinguishing CCA from HCC.

Recent evidence from our group further reinforces this concept. In our latest clinical and functional study, we demonstrated that SLPI expression is markedly increased in CCA tissues compared with adjacent normal bile ducts and hepatocytes, and that high SLPI expression is significantly associated with lymphatic metastasis, implying a role in local tumor dissemination [22]. In addition, serum SLPI concentrations were found to be elevated in CCA patients compared with those with HCC, supporting its diagnostic potential for differentiating between these two hepatobiliary malignancies. Functionally, recombinant human SLPI (rhSLPI) promoted the clonogenic survival of KKU-100 cells in a dose-dependent manner, providing direct evidence that SLPI enhances CCA cell proliferation and survival [22]. These results are in line with our present findings showing that SLPI overexpression increases tumorigenic and metastatic phenotypes, further consolidating SLPI's role as both a diagnostic biomarker and a biological driver of cholangiocarcinogenesis.

Our results also demonstrate that SLPI expression is inducible by inflammatory stimuli, particularly interleukin-6 (IL-6), a cytokine associated with chronic *Opisthorchis viverrini* infection and inflammation-induced carcinogenesis [23]. The IL-6 concentrations and exposure times used in this study were chosen to reflect sustained inflammatory signaling rather than acute cytokine stimulation. Elevated IL-6 levels have been consistently reported in the serum and tumor microenvironment of patients with cholangiocarcinoma, particularly in inflammation-associated disease settings. The observed dose- and time-dependent induction of NF-κB activation and SLPI expression at 12 and 24 h therefore supports the physiological relevance of our in vitro inflammation model. IL-6 stimulation enhanced SLPI expression and promoted tumorigenic phenotypes in both non-tumorigenic cholangiocytes (MMNK-1) and CCA cells (KKU-213A and KKU-213AL5), with a more pronounced effect observed in MMNK-1 cells. This suggests that inflammation-induced SLPI upregulation may represent an early event in cholangiocarcinogenesis, consistent with our recent findings that SLPI levels also tend to increase in *O. viverrini*-infected individuals relative to healthy controls [22]. Together, these data imply that chronic inflammation may trigger SLPI induction, predisposing bile duct epithelial cells to malignant transformation. While IL-6 stimulation was used to model inflammation-driven SLPI induction, SLPI overexpression enabled us to directly interrogate the downstream tumorigenic functions of SLPI, independent of upstream cytokine signaling.

To clarify the functional role of SLPI, we generated SLPI-overexpressing cholangiocytes. SLPI overexpression enhanced cell proliferation, migration, and vasculogenic mimicry formation (VMF), supporting its involvement at multiple stages of cholangiocarcinogenesis (CCG). These oncogenic effects are consistent with SLPI-mediated tumorigenic processes reported in other cancers [24]. Notably, SLPI overexpression in MMNK-1 cells, which are typically non-tumorigenic in vivo [25], conferred tumorigenic potential and increased metastatic capabilities. These findings suggest that SLPI can act as a transforming factor that converts normal cholangiocytes into malignant-like cells, an effect further supported by our previous observation that exogenous SLPI alone was sufficient to enhance clonogenic survival in CCA cells [22].

Beyond the IL-6→NF-κB-driven induction of SLPI shown here, several lines of evidence help rationalize how SLPI may amplify malignant traits in CCA. First, SLPI has been linked to STAT3 activation and downstream Cyclin D1 upregulation, thereby sustaining proliferation [26–28]. Second, SLPI can interact with RB/FOXM1 circuitry to promote cell-cycle progression and transcription of MMP genes, consistent with our observation of increased MMP-2/-9 activities and enhanced invasion in SLPI-overexpressing cells [29–31]. Third, tumor-stroma crosstalk may converge on PI3K/AKT signaling, as SLPI delivered in extracellular vesicles from cancer-associated fibroblasts activates this pathway in other tumors [19], suggesting a plausible axis to test in CCA. Finally, our VM data (↑VE-cadherin/VEGFA,↓N-cadherin) align with reports that SLPI promotes vasculogenic mimicry in aggressive cancers [32–33]. Together, these pathways provide a mechanistic framework in which inflammation-induced SLPI primes cholangiocytes and CCA cells to proliferate, remodel the extracellular matrix via MMPs, and adopt VM-competent states that support perfusion independently of classical angiogenesis.

Importantly, SLPI may function as a molecular bridge linking inflammation-driven signaling to vasculogenic mimicry (VM) in cholangiocarcinoma. Chronic inflammatory stimuli, particularly IL-6–mediated NF-κB activation, induce SLPI expression in cholangiocytes and CCA cells, as demonstrated in our in vitro inflammation model. Beyond its canonical role as a serine protease inhibitor, accumulating evidence indicates that SLPI can act as a signaling modulator that promotes tumor cell plasticity, survival, and extracellular matrix remodeling.

In this context, SLPI overexpression was associated with increased MMP-9 activity, upregulation of VEGFA and VE-cadherin, and suppression of N-cadherin—molecular features that are characteristic of VM-competent tumor cells rather than endothelial-dependent angiogenesis. Previous studies have further reported that SLPI can activate onco-genic pathways such as STAT3, RB/FOXM1, and PI3K/AKT, which are known to regulate transcriptional programs involved in invasion, cell fate plasticity, and VM formation. Collectively, these findings support a mechanistic continuum in which inflammation-induced SLPI integrates cytokine signaling with downstream transcriptional and proteolytic pro-grams, thereby enabling CCA cells to acquire endothelial-like properties and form VM structures independent of classical angiogenesis.

Taken together, our findings support SLPI as a clinically relevant biomarker associated with cholangiocarcinoma pro-gression. The consistent upregulation of SLPI across human transcriptomic data, animal models, and cell-based sys-tems, together with prior validation in patient tissues and sera, highlights its potential diagnostic and prognostic utility. In addition, the tumor-promoting functions of SLPI demonstrated in this study suggest that SLPI may represent a candidate therapeutic target. Nevertheless, direct evaluation of SLPI-targeted interventions was beyond the scope of the present work, and future studies will be required to determine whether SLPI inhibition can be safely and effectively exploited for therapeutic benefit in cholangiocarcinoma.

Although angiogenesis is a hallmark of cancer progression [34–35], our findings suggest that SLPI does not promote classical angiogenesis in endothelial cells. Conditioned medium (CM) from SLPI-overexpressing cholangiocytes failed to stimulate proliferation, migration, or tube formation in EA.hy926 endothelial cells, consistent with prior reports showing that SLPI may suppress fibroblast growth factor (FGF)-mediated angiogenic signaling [36–37]. However, SLPI significantly promoted VMF, characterized by increased VE-cadherin and VEGFA expression and reduced N-cadherin levels. These findings corroborate studies indicating that SLPI enhances VMF in various cancer types [38–39]. Importantly, the degree of SLPI-driven CCG was greater in CCA cells with high metastatic potential than in SLPI-overexpressing cholangiocytes, suggesting a dose-dependent role of SLPI in promoting cancer aggressiveness. Furthermore, SLPI's ability to modulate anti-apoptotic pathways may contribute to cancer cell survival under stress conditions [38–40].

This study has several limitations. (i) Some sample sizes were modest (e.g., serum and functional assays), which may limit statistical power for subgroup analyses. (ii) In the xenograft model, tumor weight differed whereas volume did not, warranting replication with longer follow-up and additional CCA models. The discrepancy observed between tumor weight and caliper-derived tumor volume may reflect limitations of geometric volume estimation, which does not account for intra-tumoral heterogeneity, necrosis, or stromal content. Direct tumor weight measurement at the experimental endpoint may better reflect total tumor burden. (iii) We mainly employed gain-of-function in MMNK-1; although informative, this does not recapitulate the genomic complexity of patient tumors. Loss-of-function (siRNA/CRISPR) and neutralization approaches for SLPI were not performed and would strengthen causal interference. Nevertheless, several experimental features support the specificity of SLPI-mediated effects observed in this study. Two independent SLPI-overexpressing clones exhibited consistent tumorigenic, metastatic, MMP activation, and vasculogenic mimicry–associated phenotypes, reducing the likelihood of clone-specific or off-target artifacts. In addition, these phenotypes closely resembled those observed in cholangiocarcinoma cell lines with endogenously high SLPI expression. Moreover, physiological induction of SLPI by IL-6 stimulation recapitulated key aspects of the malignant phenotypes, providing an independent, inflammation-driven line of evidence linking SLPI to disease progression. (iv) The use of different animal models represents both a strength and a lim-itation of this study. The hamster model was employed to recapitulate inflammation-associated cholangiocarcinogenesis

driven by *Opisthorchis viverrini* infection, whereas the mouse xenograft model was used to assess the tumorigenic consequences of SLPI overexpression in a controlled in vivo setting. Although interspecies differences in immune context and tumor microenvironment are unavoidable, the consistent tumor-promoting role of SLPI observed across hamster tissues, mouse xenografts, human cell lines, and clinical transcriptomic data suggests that SLPI-mediated oncogenic effects are conserved and biologically relevant. (v) The angiogenesis read-outs relied on EA.hy926 and conditioned media; broader endothelial contexts and in vivo angiogenesis/VM imaging would refine conclusions. (vi) In addition, survival analyses were based on publicly available transcriptomic datasets that do not provide sufficient clinicopathological annotation for multivariate Cox regression analysis. As such, whether SLPI represents an independent prognostic factor beyond established clinical variables could not be determined in this study and should be addressed in future investigations using well-characterized patient cohorts.

Future studies should investigate SLPI loss-of-function (siRNA/CRISPR) and neutralization strategies in CCA cell lines and organoids, combined with rescue experiments to confirm specificity. Mechanistic work should examine whether SLPI-driven phenotypes are dependent on STAT3, RB–FOXM1, or PI3K/AKT signaling. Additionally, interactions between CCA cells and cancer-associated fibroblast-derived vesicles carrying SLPI merit exploration. Investigation of VE-cadherin signaling in SLPI-induced vasculogenic mimicry using 3D and in vivo models would provide greater mechanistic clarity. Clinically, expanded multicenter studies should validate SLPI expression in serum and tissue samples-including *O. viverrini*-infected non-malignant cohorts-to establish diagnostic thresholds distinguishing CCA from HCC. Finally, evaluation of therapeutic combinations such as gemcitabine or targeted inhibitors with SLPI blockade could reveal synergistic effects against CCA progression.

## Conclusion

Collectively, our data underscore SLPI as a multifunctional molecule involved in both early-stage and progressive cholangiocarcinogenesis through inflammation-associated, proliferative, and vasculogenic mechanisms. Our study identifies SLPI as a critical promoter of cholangiocarcinogenesis through its involvement in cell proliferation, metastasis, and vasculogenic mimicry, rather than classical angiogenesis as illustrated in Fig 7. Together with our recent clinical findings

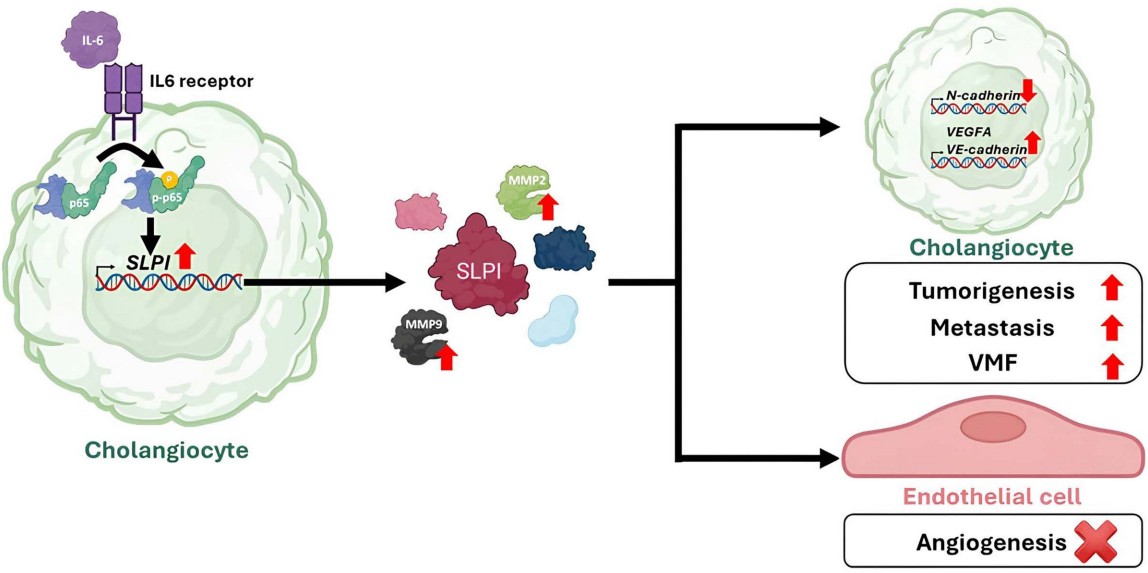

**Fig 7. The proposed model illustrates the role of SLPI on CCG.**

[22], the evidence supports SLPI as not only a promising diagnostic biomarker distinguishing CCA from HCC, but also a potential therapeutic target for mitigating inflammation-driven tumor progression in cholangiocarcinoma.

## Supporting information

**S1 Fig. Original western blot images showing the effects of IL-6 on p-p65 and SLPI expression in MMNK-1 cells (corresponding to Fig 2a).**
(TIFF)

**S2 Fig. Original western blot images showing the effects of IL-6 on p-p65 and SLPI expression in KKU-213A cells (corresponding to Fig 2b).**
(TIFF)

**S3 Fig. Original western blot images showing SLPI expression in cholangiocytes (MMNK-1), Mock control, SLPI-overexpressing cholangiocytes [SLPI(1) and SLPI(2)], early-stage CCA cells (KKU-213A), and metastatic-stage CCA cells (KKU-213AL5) (corresponding to Fig 3a).**
(TIFF)

**S4 Fig. Original gelatin zymography images showing MMP-2 and MMP-9 activity in Mock, SLPI-overexpressing [SLPI(1) and SLPI(2)], early-stage CCA (KKU-213A), and metastatic-stage CCA (KKU-213AL5) cells (corresponding to Fig 5f-g).**
(TIFF)

**S1 Table. List of primary antibodies used for IHC and western blot analyses.**
(TIFF)

**S2 Table. Primer sequences used for RT-qPCR analysis, including target genes and reference gene (GAPDH).**
(TIFF)

## Acknowledgments

We would like to express our sincere gratitude to Prof. Dr. Sopit Wongkham for her valuable guidance, constructive suggestions, and continuous encouragement throughout the course of this research.

## Author contributions

**Conceptualization:** Worasak Kaewkong, Suchada Phimsen.

**Data curation:** Jeranan Inpad, Damratsamon Surangkul.

**Formal analysis:** Kangsadan Chueajedton, Chaiwat Chueaiphuk.

**Funding acquisition:** Kangsadan Chueajedton, Suchada Phimsen.

**Investigation:** Kangsadan Chueajedton, Chaiwat Chueaiphuk, Jeranan Inpad.

**Methodology:** Kangsadan Chueajedton, Chaiwat Chueaiphuk, Damratsamon Surangkul, Suchada Phimsen.

**Project administration:** Suchada Phimsen.

**Resources:** Sarawut Kumphune, Kanlayanee Sawanyawisuth, Ubon Cha'on.

**Supervision:** Sarawut Kumphune, Suchada Phimsen.

**Validation:** Kanlayanee Sawanyawisuth, Ubon Cha'on.

**Visualization:** Worasak Kaewkong.

**Writing – original draft:** Kangsadan Chueajedton, Chaiwat Chueaiphuk.

**Writing – review & editing:** Kangsadan Chueajedton, Chaiwat Chueaiphuk, Jeranan Inpad, Sarawut Kumphune, Worasak Kaewkong, Damratsamon Surangkul, Kanlayanee Sawanyawisuth, Ubon Cha'on, Suchada Phimsen.

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
