## [Decision Letter · Decision Letter 0]

11 Nov 2025

Dear Dr. Phimsen,

Please submit your revised manuscript by Dec 26 2025 11:59PM. If you will need more time than this to complete your revisions, please reply to this message or contact the journal office at plosone@plos.org . A rebuttal letter that responds to each point raised by the academic editor and reviewer(s). You should upload this letter as a separate file labeled 'Response to Reviewers'.A marked-up copy of your manuscript that highlights changes made to the original version. You should upload this as a separate file labeled 'Revised Manuscript with Track Changes'.An unmarked version of your revised paper without tracked changes. You should upload this as a separate file labeled 'Manuscript'.

We look forward to receiving your revised manuscript.

Kind regards,

Zu Ye, Ph.D.

Academic Editor

PLOS ONE

Journal Requirements:

2. To comply with PLOS One submissions requirements, in your Methods section, please provide additional information regarding the experiments involving animals and ensure you have included details on (1) methods of sacrifice, (2) methods of anesthesia and/or analgesia, and (3) efforts to alleviate suffering.

“National Research Council of Thailand”

5. Please note that funding information should not appear in the Acknowledgments section or other areas of your manuscript. We will only publish funding information present in the Funding Statement section of the online submission form. Please remove any funding-related text from the manuscript.

**Additional Editor Comments:**

Please find attached the detailed comments from the reviewers. We kindly ask you to carefully address each point raised in your revision. When submitting the revised manuscript, please also provide a point-by-point response to the reviewers’ comments, outlining the changes made or explaining your reasoning if any suggestions were not incorporated.

Reviewers' comments:

Reviewer's Responses to Questions

**Comments to the Author**

1. Is the manuscript technically sound, and do the data support the conclusions?

Reviewer #1: Partly

Reviewer #2: Partly

Reviewer #3: Yes

2. Has the statistical analysis been performed appropriately and rigorously?

Reviewer #1: Yes

Reviewer #2: No

Reviewer #3: Yes

3. Have the authors made all data underlying the findings in their manuscript fully available?

Reviewer #1: Yes

Reviewer #2: Yes

Reviewer #3: Yes

4. Is the manuscript presented in an intelligible fashion and written in standard English?

Reviewer #1: No

Reviewer #2: Yes

Reviewer #3: No

Reviewer #1: Dear editor in chief

Chueajedton et al described Secretory leukocyte protease inhibitor (SLPI) promotes cholangiocarcinoma progression via inflammation-associated and vasculogenic mechanisms. The general concept is interesting and just some minor changes should be made to improve the manuscript.

Could the authors elaborate on how SLPI mechanistically links inflammation to vasculogenic mimicry in cholangiocarcinoma? The transition between these two roles seems conceptually broad.

Were any control experiments performed to rule out off-target effects of SLPI overexpression, especially in the context of MMP activation and VM formation?

The manuscript mentions IL-6 stimulation increasing SLPI expression. Was the IL-6 concentration physiologically relevant, and were time-course experiments conducted?

Given the use of both hamster and mouse models, how do the authors reconcile interspecies differences in SLPI expression and tumor microenvironment?

The authors state that SLPI-overexpressing cells did not affect angiogenesis but promoted VM. Was endothelial tube formation quantified, and how was VM distinguished from true angiogenesis?

The bioinformatic correlation between SLPI and poor survival is compelling. Were multivariate analyses performed to control for confounding clinical variables?

Do the authors envision SLPI as a diagnostic biomarker, therapeutic target, or both? Has its expression been validated in patient biopsies beyond bioinformatics?

Language and Structure

While the manuscript is scientifically sound, some sections (e.g., Methods and Results) could benefit from clearer transitions and more concise phrasing. Would the editor consider recommending language polishing?

Reviewer #2: This manuscript investigates the role of SLPI in cholangiocarcinoma progression. The authors effectively combine clinical data (human tissues and bioinformatics), an in vivo hamster model, and a range of in vitro functional assays to build a strong case for SLPI as a promoter of tumorigenesis, metastasis, and vasculogenic mimicry The study addresses an important knowledge gap in understanding inflammation-associated cholangiocarcinoma progression. However, several methodological details and statistical analyses need clarification, and figure presentation must be substantially improved.

The authors should explain the tumor weight/volume discrepancy in Fig 4c-e.

In vasculogenic mimicry assay (Fig 6d), the result is dramatic (279 vs. 46 structures) but the methodology is vague. How were these "tubular structures" defined and quantified? A clear description of the quantification method (e.g., number of tubes, junctions, total tube length using software like ImageJ) is necessary for reproducibility.

For statistical analysis, the description is adequate, but it should be explicitly stated whether the data met the assumptions of the tests used (e.g., normality for t-test/ANOVA). The use of "n" should be clarified for each experiment (e.g., n = 3 independent experiments, n=9 mice per group). Are “n” "three independent experiments" vs. "3 technical replicates from one experiment"? Post-hoc test specifications for ANOVA are missing for few results.

Figures are of very low resolution. Figures must be uploaded in high resolution.

Reviewer #3: Kangsadan et al in the manuscript titled “Secretory leukocyte protease inhibitor (SLPI) promotes cholangiocarcinoma progression via inflammation-associated and vasculogenic mechanisms” have attempted to understand the role of SLPI in cholangiocarcinoma.

Comments:

The role of SLPI in cancer has been extensively studied which is also mentioned by the authors, so the work adds to the knowledge database showing its relevance in cholangiocarcinoma as well. Although a variety of methods have been adopted, the manuscript lacks overall clarity.

The experimental workflow or the methodology section needs to be reorganized, adding more details on each assay, the cell line lines used etc. to better understand the sequence of experimental design.

The study also lacks loss of function experiments to validate the obtained results but this has been mentioned by the authors as a limitation of the study.

Line 103: Provide details of the total number of mice and how they were grouped and allotted for each analysis.

Line 132, 139,144: For cell culture experiments, although initially three cell lines have been mentioned, it is unclear which cells were used for each analysis and there are no details on the different groups or how the treatment was done. Please include details of this in the methodology section.

Line 139, 144, 168, 174: Wherever cell count is mentioned express them as cells /ml or cell /cm2 or cell/well.

Line 145: How were the cytosolic proteins obtained for ELISA assay.

Line 105: The number of mice, animal grouping with respective treatment needs to be mentioned.

Line 164: Was 5 mg/ml, the final MTT concentration in wells?

Figure 1a-b: Include MW markers

Figure 2c-f: It may be more appropriate to represent the Y axis of western blot graphs in this format relative expression (fold change) of pNF-κB

Fig 5f: Provide the molecular weight of zymogram bands

Line 257: If IL6 treatment can upregulate SLPI expression and activate pathway, why was it not preferred SLPI over overexpression in studies investigating its function.

**Do you want your identity to be public for this peer review?** For information about this choice, including consent withdrawal, please see our Privacy Policy

Reviewer #1: **Yes:** Shadi Aghamohammad

Reviewer #2: No

Reviewer #3: No

---

## [Author Response · Author response to Decision Letter 1]

17 Dec 2025

Dear Editorial Office,

We have carefully revised the manuscript in response to all reviewers’ and editorial comments. All issues raised have been addressed point by point in the accompanying Response to Reviewers document, and the manuscript has been revised accordingly with tracked changes.

In addition, all figures have been replaced with high-resolution versions that comply with the journal’s technical requirements.

We sincerely appreciate the reviewers’ and editors’ constructive comments, which have helped to improve the clarity and quality of our manuscript.

Sincerely,

Suchada Phimsen, Ph.D.

---

## [Decision Letter · Decision Letter 1]

29 Dec 2025

Secretory leukocyte protease inhibitor (SLPI) promotes cholangiocarcinoma progression via inflammation-associated and vasculogenic mechanisms

PONE-D-25-54451R1

Dear Dr. Phimsen,

We’re pleased to inform you that your manuscript has been judged scientifically suitable for publication and will be formally accepted for publication once it meets all outstanding technical requirements.

Kind regards,

Zu Ye, Ph.D.

Academic Editor

PLOS One

Additional Editor Comments (optional):

Reviewers' comments:

Reviewer's Responses to Questions

**Comments to the Author**

Reviewer #1: All comments have been addressed

Reviewer #2: All comments have been addressed

2. Is the manuscript technically sound, and do the data support the conclusions?

Reviewer #1: Yes

Reviewer #2: Partly

3. Has the statistical analysis been performed appropriately and rigorously?

Reviewer #1: Yes

Reviewer #2: I Don't Know

4. Have the authors made all data underlying the findings in their manuscript fully available?

Reviewer #1: Yes

Reviewer #2: Yes

5. Is the manuscript presented in an intelligible fashion and written in standard English?

Reviewer #1: Yes

Reviewer #2: Yes

Reviewer #1: The authors are fully addressed the comments and the manuscript is completely improved. the manuscript is now appropriate to be published.

Reviewer #2: The authors have tried to properly address the reviewer's comments. The manuscript can be accepted with these revisions.

**Do you want your identity to be public for this peer review?** For information about this choice, including consent withdrawal, please see our Privacy Policy

Reviewer #1: **Yes:** Shadi Aghamohammad

Reviewer #2: No

---

## [Editor Report · Acceptance letter]

PONE-D-25-54451R1

PLOS One

Dear Dr. Phimsen,

I'm pleased to inform you that your manuscript has been deemed suitable for publication in PLOS One. Congratulations! Your manuscript is now being handed over to our production team.

Kind regards,

on behalf of

Prof. Zu Ye

Academic Editor

PLOS One